# Caring for Older People during and beyond the COVID-19 Pandemic: Experiences of Residential Health Care Workers

**DOI:** 10.3390/ijerph192215287

**Published:** 2022-11-19

**Authors:** Veronica Sze-Ki Lai, Sui-Yu Yau, Linda Yin-King Lee, Becky Siu-Yin Li, Susan Sin-Ping Law, Shixin Huang

**Affiliations:** 1School of Nursing and Health Studies, Hong Kong Metropolitan University, Hong Kong, China; 2Department of Sociology and Social Policy, Lingnan University, Hong Kong, China

**Keywords:** COVID-19, long-term care facilities, health care workers, delivery of care, qualitative study

## Abstract

Older people and health care workers in residential care homes are particularly vulnerable to the adverse impacts of the COVID-19 pandemic. As COVID-19 has been spreading around the world for more than two years, the nature of care delivery has been substantially transformed. This study aims at understanding the long-term and ongoing impacts of COVID-19 on the delivery of care in residential care homes. It investigates how the delivery of care has been transformed by the COVID-19 pandemic and how health care workers adapted to these changes from the perspectives of frontline health care workers. Semi-structured interviews were conducted from February to December 2021 with a purposive sample of 30 health care workers from six residential care homes in Hong Kong. Thematic analysis identified three themes, including (1) enhancing infection prevention and control measures; (2) maintaining the psychosocial wellbeing of residents; and (3) developing resilience. Discussions and implications were drawn from these findings.

## 1. Introduction

The global trends of population aging and the rise in life expectancy have significantly increased the need for long-term care (LTC) services, especially for those aged 80 years and above [1]. Residential care homes provide essential LTC services to address the health, personal care, and social needs of a vulnerable population of frail older people who experience significant declines in their intrinsic capacity and functional ability [2]. The COVID-19 crisis, however, has put residential care homes at risk. COVID-19 has disproportionately affected older people and health care workers in residential care homes. Older people in LTC facilities are more likely to have compromised immune systems or chronic conditions, and they are exposed to a high risk of infection and mortality due to the COVID-19 pandemic. Across countries that are part of the Organization for Economic Cooperation and Development (OECD), 40% of total COVID-19 deaths have occurred in the LTC sector [3]. Staff in residential care homes are also at higher risk of catching COVID-19. An English study found that not only did health care workers have a higher risk of catching COVID-19, but they also had a more than seven-fold higher risk of contracting severe COVID-19 compared to non-essential workers [4].

Residential care homes are often characterized by high population densities, shared rooms, and frequent person-to-person contact in the delivery of personal care, which are all facilitators of the transmission of COVID-19 [5]. Other organization-level characteristics, such as larger sizes, higher occupancy rates, being a for-profit facility, older building age, and the absence of pre-existing infection prevention and control (IPC) protocols, were also found to be associated with adverse outcomes of COVID-19 in residential care homes [6,7]. It is thus pivotal to develop strategies tackling the needs of residents and health care workers in residential care homes, to prevent and manage the impacts of COVID-19 [8].

The detrimental impacts of the COVID-19 pandemic on residential aged care and its care workforce have been widely evaluated. This unprecedented public health crisis has led to widespread outbreaks of infection and mortality among residential care home residents and health care workers [9]. The crisis has also created complex and stressful circumstances for health care workers, especially at the onset of the pandemic. Research has reported that care workers in residential care homes faced a variety of challenges at the beginning of the pandemic, including constraints on personal protective equipment (PPE) and testing, as well as fear and stress associated with the risk of infecting themselves and their families [10]. The prolonged isolation and social restrictions put in place to curtail the transmission of the virus led to declining psychological wellbeing and cognition, as well as increased problem behaviors among residents, which created further strains related to care for residential care home staff who provided everyday care to residents [11,12]. In addition, health care workers in residential care homes have faced substantial difficulties related to reduced staffing levels and increased workloads [13]. Under such challenging circumstances, health care workers in residential care homes have reported experiencing increased stress, anxiety, and depressive symptoms, as well as physical and emotional burnout [14].

The vulnerability experienced by residential care home health care workers during the pandemic has occurred within a variety of pre-existing structural problems related to care delivery in the aged care sector. One of the most widely debated issues is the workforce shortage crisis in the aged care sector. For a long time, residential health care workers have been working under precarious work conditions characterized by part-time and multiple employment, heavy workloads, low wages, and stigmatized social images [15]. These unfavorable conditions have led to chronic understaffing in residential care homes globally, a problem that has been exacerbated by the COVID-19 pandemic [14,16].

Despite its detrimental impacts, the COVID-19 pandemic has provided an opportunity for residential care home settings to reflect on and address problems regarding care delivery in the long run. For example, Duan et al. [17] examined the multi-site work arrangements of health care aides during the COVID-19 pandemic in Canada. They suggested that this staffing assignment, a practice resulting from the long-term care workforce shortage, became unsustainable during the pandemic because of the high risks of COVID-19 spreading and long-term initiatives should be adopted to ensure stable staffing and continuity in care. Gunawan et al. [18] pointed out that, aside from its negative impacts, the COVID-19 pandemic also facilitated innovation and creativity in nursing care delivery, especially through the increased use of telehealth technologies. A study examining nurses’ occupational satisfaction during the COVID-19 pandemic also revealed that, despite operating under difficult circumstances, nurses, particularly those working in the community, such as in residential care homes, did not experience a significant decrease in occupational satisfaction; instead, they were highly motivated by occupational values [19].

Even though fruitful discussions have been developed to advance our understanding of the impacts of the COVID-19 pandemic on residential care homes, most existing studies have focused on the early outbreak of the COVID-19 pandemic, when the widespread transmission of an unknown disease led to a state of public health emergency and devastating outcomes in residential care homes. The ongoing, long-term impacts of the pandemic have thus remained under-evaluated. Moreover, discussions in current studies have focused primarily on the health and wellbeing outcomes of health care workers and older people during the COVID-19 pandemic. Relatively limited attention has been paid to how everyday care delivery in residential care homes has been transformed and adapted during the COVID-19 pandemic. As COVID-19 has been spreading around the world for more than two years, the nature of care delivery has been substantially transformed, given that the need to perform IPC, as well as the consequences associated with COVID-19 containment measures, has become a “new normal” in the everyday operations of residential care homes. It is thus imperative to understand how COVID-19 has transformed the delivery of care, the health care workforce, and residents’ wellbeing in residential care homes in the long run. Moreover, current research has predominantly adopted quantitative or mixed methods to evaluate the impacts of the COVID-19 pandemic on residential care homes. Because of the limitations caused by the emergency of the outbreak and social distancing restrictions, there has been only a scarce amount of in-depth qualitative research investigating the impacts of the pandemic on the delivery of care in residential care homes from the perspectives of frontline health care worker [20,21,22].

Using the case of Hong Kong, this qualitative study aims to examine the long-term impacts of COVID-19 on residential care homes from the perspectives of frontline health care workers. This study addresses the following questions: (1) What do health care workers perceive as the impacts of the persistent circulation of COVID-19 on their delivery of care in residential care homes? (2) What are health care workers’ experiences of coping with and adjusting to the changing modes of care delivery in residential care homes? (3) From the perspectives of frontline health care workers, what are the consequences of these changing modes of care delivery for health care workers and residents?

The case of Hong Kong provides a unique opportunity to examine the long-term impacts of the COVID-19 pandemic on the LTC setting because of its unique demographic characteristics and approach to the COVID-19 response. Hong Kong has become a “super-aging” society, characterized by population aging, long life expectancies, and low levels of fertility [23]. Moreover, Hong Kong has one of the highest institutionalization rates of older people among developed economies, with about 8% of people over 70 years old living in residential care homes [24]. These demographic characteristics have made the city particularly vulnerable to the risks of adverse COVID-19 outcomes. Despite these risk factors, before the fifth wave of the COVID-19 outbreak dominated by the Omicron variant, Hong Kong had recorded relatively low levels of infection and mortality in residential care homes. Falling in line with the zero-COVID policy pursued by the Chinese central government, Hong Kong reported fewer than 4000 COVID-19 cases in 2021 [25]. The low community transmission level kept residential care homes and their residents relatively safe from infection.

The effective early COVID-19 response in Hong Kong was partly due to the lessons learned from the severe acute respiratory syndrome (SARS) epidemic in 2003 [26]. Following the SARS epidemic, the Hong Kong government set up policies and protocols related to IPC in residential care homes, including publishing the first Guidelines on Prevention of Communicable Diseases in Residential Care Homes for the Elderly in 2004 and setting up Infection Control Officers in all residential care homes to coordinate and implement IPC policies and measures. The SARS epidemic also significantly raised public health awareness and knowledge among citizens and health care workers in Hong Kong, as marked by popular adherence to behaviors such as mask wearing, hand hygiene, and social distancing during the pandemic [27].

## 2. Materials and Methods

### 2.1. Study Design

This study adopted a qualitative descriptive design, an approach that is particularly relevant in nursing and health care research that explores participants’ experiences and perceptions [28]. This paper was produced as part of a larger research project examining health care workers’ experiences and perspectives of care work in residential care homes in Hong Kong. The 32-item Consolidated Criteria for Reporting Qualitative Research (COREQ) checklist was employed to guide the design of the study (Appendix A). To encourage reflexivity [29], authors from different disciplines and background (i.e., nursing, social work and sociology) met regular to review and discuss the design of research strategies, the progress of data collection and data analysis, so as to reflect, compare, and challenge the underlying assumptions of each other.

### 2.2. Participants and Sampling Strategy

A purposive sample of 30 health care workers was recruited. The following inclusion criteria were adopted to select research participants: (1) participants must work in one of the following roles: registered nurses (RN), enrolled nurses (EN), health workers (HW), and personal care workers (PCW); (2) participants must have at least six months of work experience in a residential care home; and (3) participants must provide frontline care services to older people in a residential care home. Exclusion criteria were: (1) care workers who adopted only managerial roles and did not provide frontline care; and (2) RCHE workers working in other roles (e.g., social workers, occupational therapists, physical therapists).

Employing the maximum variation sampling strategy, participants in different roles were purposively recruited from different residential care homes, including publicly subsidized and private facilities, to allow for heterogeneous work experiences and perspectives among health care workers working on the frontline during the COVID-19 pandemic. The six residential care homes were recruited from the authors’ professional network and the participants were recruited by the managerial staff of each residential care home, such as the nursing manager and the facility-in-charge. Participants were fully informed about the purpose and procedures of the study, as well as their rights within the study. Written informed consent was gained from both participants and residential care homes before data collection commenced.

### 2.3. Data Collection

Data were collected from February to December 2021, a period during which the pandemic was relatively under control and case levels remained low in Hong Kong because of the implementation of the zero-COVID policy by the Hong Kong government. Semi-structured, in-depth interviews were conducted with each participant. The interview questions revolved around health care workers’ work experiences before and during the pandemic, as well as their perspectives on how the COVID-19 pandemic had impacted their care work. Guiding questions included:As the COVID-19 pandemic has been ongoing for more than one year, from your experiences, what are the impacts of the pandemic on the delivery of care in your residential care home?In what ways do you think the delivery of care during the pandemic differs from that in the past?How do you and your colleagues cope with and adapt to these changes?From your perspective, what are the impacts of these changes on the quality of life of residents?

Following semi-structured interview method’s emphasis on the relational focus and flexibility in the process of data collection [30], the four questions served primarily as a flexible interview guide for interviewers in this study to explore and delve into research participants’ experiences and perspectives of care delivery during the pandemic. These questions also provide a guidance to purposively ask follow-up questions based on participants’ responses. Personal interviews were conducted in a private meeting room, arranged by the residential care home. The length of each interview was 55 min on average. Interviews were audio-recorded and transcribed verbatim with the written consent of the participants.

### 2.4. Data Analysis

The text of the interviews was inductively analyzed using thematic analysis, with the aim of achieving “an understanding of patterns of meanings from data on lived experiences” [31]. Three procedures were followed in the process of data analysis, using the qualitative data analysis software NVivo 12. First, familiarity with the data was gained through reading the interview text in an open-minded manner several times, in order to solicit novel information from the data. Then, meanings and themes related to the participants’ lived experiences were searched for, marked, compared, and organized into preliminary themes and patterns. Finally, the themes were organized into a meaningful whole [31]. More specifically, each interview transcript was independently coded by two authors (V. Lai and S. Huang), who are experienced qualitative researchers in the interdisciplinary fields of nursing (V. Lai) and social work (S. Huang). The two authors met regularly to review and discuss codes and their meanings. Disagreements were discussed and addressed until a consensus was met between the two coders. Then, a coding frame was generated and presented to a third author (I. Yau), who is a qualitative researcher of geriatric nursing, for review before applying systematically to all data. Two authors (V. Lai and S. Huang) then independently categorized these codes into themes and came together again to discuss, compare, and eventually meet agreements about the themes and sub-themes that meaningfully conceptualized health care workers’ perspectives of how the COVID-19 pandemic had transformed their care delivery. The themes, sub-themes, and coding frame were then presented and debriefed to all authors for peer review, and team discussions were held to further amend the conceptualizations.

## 3. Results

A total of 30 health care workers from six residential care homes participated in this study (Table 1 and Table 2). Three themes and ten sub-themes (Table 3) were identified as the long-term impacts of the COVID-19 pandemic on the delivery of care in residential aged care homes. The three themes were (1) enhancing infection prevention and control measures; (2) maintaining the psychosocial wellbeing of residents; and (3) developing resilience.

### 3.1. Enhancing Infection Prevention and Control Measures

#### 3.1.1. Incorporating IPC Measures in Daily Care Routines

Health care workers reported that their care work in residential care homes had transformed substantially during the pandemic, with IPC measures becoming an indispensable part of their daily work. As revealed by the participants, although infection control has always been a part of residential care, the COVID-19 pandemic has made IPC measures more important and ubiquitous than since before the SARS epidemic in 2003. These IPC measures have significantly transformed the daily care routine of health care workers, increased their workloads, and imposed new challenges to the delivery of care in residential care homes.

Health care workers reported that IPC measures, including maintaining the personal hygiene of both themselves and the residents, maintaining social distancing, checking temperature, detecting early symptoms, isolating residents discharged from hospitals, and conducting regular testing, have become part of their daily routine since the beginning of the pandemic:


*“There are many more infection control measures in place now. This was especially the case at the onset of the pandemic. I did everything I could, such as changing gloves frequently, washing my hands, and taking my temperature because I worried I might infect the residents.”*
(Participant 20, PCW)


*“We need to shower the residents after they come back from follow-ups in the hospital. That means it will increase our workload. Besides showers, we have to keep observing their body temperature for two more weeks after they come back.”*
(Participant 24, health worker)

While the implementation of IPC measures had now been directly integrated into the daily care routine of health care workers, these preventive measures also unintentionally and indirectly affected other aspects of care delivery in residential care homes. This is particularly the case of the implementation of the visiting restriction policy. In the face of the prolonged visiting restriction policy, health care workers reported that they needed to undertake additional care workloads that used to be shared by family caregivers.


*“It is much busier for PCW staff because family members or domestic helpers of the residents are not able to help us now. Before the restriction, they will help with feeding, changing napkins, blow-drying hair, or assisting care workers with some daily activities for the residents.”*
(Participant 22, health worker)

As suggested by this research participant and many others, compared to HWs and nurses, PCWs’ care workloads were particularly affected by the IPC measures in the pandemic. Participants revealed that, prior to the pandemic, many family caregivers, particularly residents’ adult children and domestic helps, often time paid frequent, even daily, visits to older people in residential care homes and took some of the direct care activities such as feeding as gestures of care and support. Since family caregivers had been restricted in paying frequent visits to older people and thus were unable to share the direct care activities, PCWs had to undertake all those care tasks, which imposed further strains on their already overloaded jobs. The impacts of increased workloads could be particularly stressful for health care workers in private residential care homes, whose health care workers-to-beds ratio was often lower than that of contract and public homes that were partially or fully subsidized.

Although health care workers perceived IPC measures to have been pivotal in maintaining the safety of residents and themselves, they revealed that the development and implementation of general IPC measures often lacked appropriateness when being applied to the aged care setting. A health care worker, for example, pointed out the limitation of the testing program for residents:


*“We must perform regular testing on our residents now…We collect deep throat saliva specimens for testing regularly according to the regulations of the government... It is very easy for normal people to spit. However, when I started doing it, I instantly found it more difficult than collecting bird’s saliva! It turned out that older people secrete very little saliva and can hardly spit anything. Even though I have every incentive to perform testing because it is important for protecting our residents and ourselves, I cannot do it properly.”*
(Participant 17, enrolled nurse)

As suggested by this example, the inability of public health policies to take into account the conditions of older people and the residential care home setting significantly compromised the effectiveness of IPC measures. It also led to poor compliance with IPC policies and measures among health care workers in some cases, which might further aggregate the residents’ risks of infection.

#### 3.1.2. Performing Health Education and Promotion

Health care workers reported they were expending effort to promote long-term adherence to health behavior through the health education of residents and their family caregivers during the pandemic. One example health care workers frequently mentioned in the interviews was promoting older adults’ adherence to wearing a mask. Participants reported that, in order to prevent the spread of COVID-19, residents were advised to wear masks in public areas in their residential care homes. Health care workers described that they found it challenging to promote mask-wearing behaviors among older people at the beginning of the pandemic, especially with residents with profound dementia. Over time, however, they witnessed behavioral changes among residents:


*“I keep reminding them about wearing masks again and again. Older people sometimes wear their mask on the top of their heads and sometimes pull down their mask. So, whenever I see them like this, I remind them how to wear a mask. Now, the residents have got used to it. They even ask me for a mask if they are running out of them. It has become much easier.”*
(Participant 24, HW)

As described by the participants, older people often had limited awareness about IPC-related health behaviors prior to the COVID-19 pandemic. When IPC became a pivotal part of their work in the pandemic, health care workers stressed the importance of educating and promoting precautionary behaviors, especially mask wearing and hand hygiene, among older people to prevent the transmission of COVID-19 within the facility. They did so by exploring age-friendly communication in daily interactions with older people such as using repeated reinforcement, simplified language, and peer education to facilitate older people’s access to health information and adherence to health behaviors.

### 3.2. Maintaining the Psychosocial Wellbeing of Residents

When discussing the long-term impacts of the pandemic, most health care workers suggested that the most significant impact was on the psychosocial wellbeing of residents in residential care homes. Health care workers revealed that IPC measures, especially visiting restrictions, had disrupted the intergenerational bonding between older people and their family members and brought adverse wellbeing outcomes to older people. In order to tackle these challenges, health care workers had developed adaptive ways of sustaining the intergenerational bonding and building rapport with residents.

#### 3.2.1. Interruption of Intergenerational Bonding

As revealed by the health care workers, prior to the COVID-19 pandemic, family caregivers played a significant role in the planning and delivery of care for older people in residential care homes. They indicated that the COVID-19 pandemic and the precautionary measures accompanying it, especially the visiting restrictions of family members, have drastically interrupted family caregivers’ involvement in the planning and delivery of care for older people in residential care homes:


*“In the past, family members could stay in the institution for a whole morning. They could take their mothers or grandmothers to eat outside. The residents were much happier. Now, during the pandemic, they are less happy because they are confined [to the facility]. Even though they can have a video call, it is not the same as a face-to-face meeting. Now, family members can come and visit if they test and are vaccinated, but the visit time is still limited.”*
(Participant 29, HW)

As elaborated by the participants, the intergenerational bonding showcased through family caregivers’ care involvement, such as frequent on-site visits and home visits, has been an important part of the quality of life and social support of residents. However, since family members were no longer able to have frequent on-site visits to older people in residential care homes in the pandemic, the previous patterns of intergenerational interactions were significantly interrupted, which in turn engendered adverse psychological wellbeing outcomes among older people.

#### 3.2.2. Witnessing the Adverse Wellbeing Outcomes of Residents

Participants described that when prolonged restrictions on family visitors and home visits were imposed, older people were left in isolated and lonely circumstances. It was reported by most health care workers in this study that many residents developed mental health problems related to depression and anxiety:


*“The biggest influence is on older people. They miss their family members a lot. Older people also need psychosocial care. [The isolation] has resulted in a lot of emotional issues for them. They often have tears in their eyes. This is the influence.”*
(Participant 17, EN)

As described by the participants, older people’s emotional distress primarily stemmed from separation from family members because of the visiting restrictions. This was especially the case at the onset of the pandemic when family members were completely barred from visiting the residential care homes. Even after the visiting restriction was lifted, the inability to have frequent interactions with family members still triggered a strong sense of grievance among older people, who often believed that they were abandoned by their family members in the care homes.

In addition to the adverse impacts on older people’s mental health, health care workers also witnessed the physical and cognitive deterioration of older people in the pandemic:


*“When the residents have less and less contact with the outside world, their memories deteriorate over time. This is especially the case for older people with dementia. Their memory has worsened, and they have become unable to recognize people around them. These kinds of conditions have become more common.”*
(Participant 7, EN)

#### 3.2.3. Sustaining Intergenerational Bonding in Adaptive Ways

In the face of these challenges, as revealed by the health care workers, communicating with family caregivers and searching for alternative ways to sustain intergenerational bonding have become pivotal to the delivery of care work in residential care homes.

After observing the emotional distress of residents because of separation from family members, health care workers in our study continued to search for and experiment new ways of maintaining the intergenerational interactions between older people and their family members. Most noticeably, it is a common practice for residential care homes in our study to arrange health care workers to receive and pass on food and daily supplies from family caregivers to residents, as a new gesture of care and bonding:


*“Sometimes [the residents] miss their family members a lot and become emotional. In such circumstance, I have to work on this issue. I usually call their family members and ask them to call the residents or bring some daily supplies to the care home. This is particularly challenging in the past two years because older people’s psychological conditions are unstable in the pandemic.”*
(Participant 26, HW)

Using tangible daily supplies such as foods that were personally sent by their family members, health care workers were then able to comfort older people that they received continuous care and support from family members and thus relax their emotional distress.

Health care workers, particularly HW, EN, and RN who were responsible for communicating with family members, also reported that their communication with family members increased drastically during the COVID-19 pandemic. As described by the participants, the communication between health care workers and family members revolved primarily around updating the physical and psychological conditions of the residents:


*“I think our communication with family members increased drastically during the pandemic. Many family members came and visited their parents every day before the pandemic. They can no longer do that now and might call us every day to ask how their moms are doing or if they are in a bad mood today instead. We have to address these questions from family caregivers because I think it is part of caring for the wellbeing of their parents.”*
(Participant 22, HW)

In addition, health care workers also reported that part of their care work now shifted to facilitating the visiting arrangements of family members and coordinating video calls between residents and their family members:


*“Now the visiting restrictions have been gradually lifted, family members can visit again. [Family members] asked tons of questions about how they can visit, what the requirements are, whether or not they need to get vaccinated, etc… In the middle of the pandemic, some family members were particular worried, and we thus… facilitated video calls so they could at least have some sort of interaction with the residents.”*
(Participant 17, EN)

#### 3.2.4. Building Rapport with Residents

Moreover, health care workers reported that they devoted more efforts to maintaining the psychosocial wellbeing of the residents in their everyday care work. Participants reported that they actively built rapport with residents, listening to their grievances, talking to them, and inventing social activities within the residential care homes:


*“Because of the pandemic, [the residents] cannot see their family members for a long time. They may feel emptiness when they have less chance to see their families. Some of them have poor appetites and low moods. Some even develop depression. I take the initiative to talk with the residents when they develop these symptoms. When they are not willing to let me know what has happened, we figure something out to make them feel less lonely. We also try to plan activities for them. This has now become part of our jobs.”*
(Participant 21, EN)

As suggested by participants, while the COVID-19 pandemic has imposed detrimental impacts on residents, it has surprisingly elevated the significant role of psychosocial care in residential aged care work. Through attending to the psychosocial wellbeing of residents, health care workers reported that they have built rewarding and intimate relationships with residents:


*“During this period, I have a much closer relationship with older people. Because the residents have no family visitors, we spend more time chatting with them. When we chat more, I feel much closer and more intimate when I provide direct care to them. I want to make them feel not alone even though their family members are not here. This is the favorite part of my work.”*
(Participant 8, PCW)

### 3.3. Developing Resilience

Although the pandemic has created extraordinary challenges for health care workers in residential care homes, the participants in this study reported that they have developed resilience and been able to adapt to the transforming modes of aged care over time.

#### 3.3.1. Experiencing Stress at the Onset of the Pandemic

Health care workers recalled that, at the onset of the pandemic, they experienced a high level of psychological stress because of concerns about exposing themselves, their family members, and residents to the risks of COVID-19 infection. A participant vividly recalled her stressful experiences at the onset of the pandemic:


*“At the beginning of the pandemic, I was extremely panic, although now it is much better. At the time I worried that I might carry the virus and pass on to older people one by one. So I took full vigilance of following all IPC measures such as rubbing hands with alcohol frequently, taking temperature and everything. It was very frightening.”*
(Participant 20, PCW)

As described by the participants, the special nature of the residential care homes, in terms of the dense and shared living spaces, the vulnerability of older people in frail physical conditions, and the nature of their direct care work which entailed frequent contacts with dozens of residents, made them particularly stressful about the infection and transmission of COVID-19 in their work:


*“In the beginning, I was quite worried because our institution was quite big and busy. So, I worry [about being infected].”*
(Participant 8, PCW)


*“I worried that I might bring the virus from my daily life to the residents. You know, it will affect a lot of people. So, every day when I was seeing so many residents, I worried that I might accidentally become a transmitter.”*
(Participant 17, EN)

Participants also suggested that the stresses and challenges they experienced at the beginning of the pandemic came from the lack of clear policy guidelines and support from the government, as well as the acute shortage of workers during the pandemic:


*“Colleagues called back and asked if they should go back to work if their neighbor was infected by COVID-19. It is difficult to make decisions in this situation, with a lack of support and guidelines from the government. Sometimes we ask them to stay home to avoid the risk of infection for our residents and we have to take on an extra workload.”*
(Participant 4, RN)

#### 3.3.2. Seeking Resilience in Adverse Circumstances

Despite facing stress and challenges, health care workers developed resilience in adverse circumstances. For example, some participants explained that they had been adapting to the IPC measures and improvised new modes of care provision over time:


*“We were stressful but soon became happy again. We have held many activities. We held the Lunar New Year fair. We sought happiness in our small world. We held Christmas parties by grouping four persons per table for both staff and residents [to abide by the social distancing rule]. It is a challenge for us to figure out how to cope with adversity.”*
(Participant 18, RN)

Health care workers indicated that innovating these new modes of care provision could ensure the continuity of care and was pivotal to the quality of life of older people in the pandemic. Many participants described that, despite being constrained by IPC measures, their care work still paid off and maximized the quality of life of older people to the greatest extent possible. The positive transformations of older people in turn promoted job satisfaction among health care workers:


*“Making them happy is more important than anything else because they have already separated with their family members… When [I] witness the resident turned from unhappy to happy-the transformation can be very obvious-I feel like I do the right thing and have a great sense of achievement.”*
(Participant 27, RN)

Participants mentioned that external support from family members and other social service organizations was significantly curtailed in the pandemic. Health care workers thus saw themselves as the primary, if not the only, source of support for older people. This capacity to continue to care, support, and bring positive changes to older people in an adverse environment also cultivated a strong sense of meaningfulness among the health care workers:


*“When I interact with older people at work, I feel very satisfied. It feels good to see that they are living comfortable and happy lives. Even though life can be boring and difficult in the institution now, I hope I can bring a little bit of happiness to the residents. This would make me happy.”*
(Participant 22, HW)

#### 3.3.3. Joining the LTC Sector in the Midst of the Pandemic

Moreover, even though the pandemic had revealed and exacerbated the long-existing problem of workforce shortages, it also surprisingly turned into an opportunity to attract new staff to the LTC sector. Several participants of this study joined the LTC sector in the middle of the pandemic because of the relative job security and career prospects the sector offered as compared to other sectors:


*“Now that we are in the middle of the pandemic, a lot of people have lost their jobs. People start thinking about money and they realize that [long-term care] is the most stable and secure sector. I have always thought that older people have a lot of care needs and have thus wanted to be a nurse in this sector. At least I won’t lose my job suddenly if I work in the aged care sector.”*
(Participant 17, EN)

As described by the participants, some of them joined residential care homes as PCWs after losing their jobs in other sectors such as the service and trade industries in the middle of the pandemic. For them, even though personal care work was physically and emotionally demanding, it remained attractive. Participants suggested that population aging had increased the societal demands for health care workers in the LTC sector, which in turn created plenty of job opportunities and provided stable income for them, particularly for those working in publicly subsidized residential care homes. For some PCWs joining the LTC sector during the pandemic, they proposed that the sector’s relatively promising career ladders that offered development opportunities for them to become HW and EN through continuous education would also incentivize them to pursue a long-term career in the sector after the pandemic.

## 4. Discussion

This study evaluated the long-term impacts of the COVID-19 pandemic on the residential aged care sector, focusing on how the delivery of care has been transformed by the COVID-19 pandemic and how health care workers adapted to these changes. Drawing on the findings of this study, several lessons can be learned to re-envision residential care in the post-pandemic world.

The findings of this study propose that IPC measures have become an indispensable part of everyday care routines in residential care homes since the onset of the pandemic and will likely be continued for a long time in the future. The effective implementation of appropriate IPC policies and measures can significantly reduce the risk of COVID-19 transmission among residents, staff, and caregivers in residential care homes [32,33]. More than two years into the pandemic, residential care homes and health care workers in Hong Kong have established a set of good practices and experiences related to delivering effective IPC precautionary measures, partly thanks to pre-existing IPC infrastructure and awareness. Consistent with existing research findings [20,34,35], this study demonstrates that the persistent implementation of IPC policies and measures during the pandemic has transformed daily care routines and increased the workloads of nurses and personal care workers. We further pointed out that such changing modes of care delivery had particularly increased the workloads of personal care workers who undertook the majority of direct care work. Future research and clinical initiatives should be carried out to develop evidence-based practices, such as technology-mediated programs, to relieve the strain on health care workers, to have extra support for direct care workers, and to promote the optimal and efficient implementation of IPC measures in the long run.

This study adds nuances to existing studies by revealing that, in order to promote the compliance and effectiveness of IPC measures, the development and implementation of IPC policies and measures need to be sensitive and appropriate to the conditions of older people. Existing studies have revealed that the COVID-19 pandemic exacerbated existing and engendered new forms of disparities among older people in terms of the access to health care and health information in the community and acute care settings [36,37,38,39]. Our study further provided evidence of the health care disparities of older adults in LTC facilities. As revealed by the current study, public health policies and measures related to IPC, such as regular deep throat saliva testing schemes, might not be applicable in LTC facilities that predominantly serve frail older people. A recent study among older adults in Asia revealed that easy access to health services is significantly associated with good COVID-19 preventive behaviors [40]. The current study provided further evidence regarding the role of health care workers in promoting health behaviors such as mask use and hand hygiene among older people in LTC facilities through age-friendly health education. The experiences of health care workers demonstrate the effectiveness of information reinforcement and simplification [41], as well as peer education, in promoting the health behaviors of residents. Future efforts should be expended to examine the promotion of age-inclusive IPC measures, which have been largely missing in existing research and clinical interventions to date.

Moreover, this study examined the implications of COVID-19 precautionary measures, especially visiting restrictions, for the psychosocial wellbeing of older people living in residential care homes from the care experiences of frontline health care workers. Our findings echo the increasing scholarly attention paid to the detrimental impacts of the pandemic and IPC measures on the psychosocial wellbeing of residents in LTC facilities [12,35,42]. Total visitor bans and visitor restrictions have not only increased the workloads of health care workers, as family caregivers can no longer share care work as they used to, but have also led to emotional distress, a sense of loneliness and isolation, and cognitive and physical deterioration among older people. This study adds to the existing literature by highlighting the cultural sensitivity in the discussion of psychological wellbeing in the pandemic. In the unique sociocultural context of Hong Kong, family caregivers’ involvement in the planning and delivery of care for older people in residential care homes is deemed to be an essential extension of filial piety commitment and a continuation of intergenerational relationships [43]. The impacts of visiting restrictions on the psychosocial wellbeing of older people should be understood as a drastic disruption of intergenerational bonding, which entails adverse sociocultural and psychological consequences within Chinese society. These findings indicate the importance of carefully balancing the tension between residents’ physical safety and psychosocial wellbeing in future responses to the pandemic. International consensus has started to recognize the urgency of the safe and flexible reopening of residential care homes through promoting vaccination, implementing safe visiting practices, and considering the community’s and facilities’ transmission status [44,45]. The findings about the detrimental impacts of COVID-19 precautionary measures on intergenerational bonding in Chinese society also indicate the importance of improvising adaptive ways to sustain intergenerational relationships as well as considering cultural sensitivity in supporting the psychosocial wellbeing of older people in LTC facilities in the future.

In addition, this study indicates that communication between health care workers and family members has become more salient in the care of older people during the COVID-19 pandemic. The prolonged visiting restrictions and other IPC measures in the pandemic have created challenges for health care workers to communicate with family members about the care needs of older people in LTC facilities. This theme of communication challenges has been repeatedly recognized not only in the LTC setting, but also in other care settings such as hospitals and schools [46,47,48,49]. While adaptive ways such as telecommunication solutions were found to be promising in the pandemic [50], communication with family members could also create anxiety and strains among care workers [51]. As further revealed by the current study, while health care workers reported that communication with family members had significantly increased their workloads in the pandemic, they generally believed that it is an important part of their care work in order to maintain the intergenerational bonding that is essential to the quality of life of older people in LTC facilities. It is thus recommended to develop clear protocols and guidance, provide institutional support and training, and innovate more technology-mediated modes of communication to facilitate the communication between health care workers and family members.

The findings of this study indicate the important role of frontline health care workers in maintaining the psychosocial wellbeing of older people in residential care homes. While existing studies have extensively examined the pandemic’s devastating psychosocial impacts on older people in LTC facilities, the current study takes a step forward to unravel how health care workers addressed this challenge in their everyday care, a theme that is yet to receive adequate attention. Health care workers have reported growing awareness of and efforts to care for the mental health of residents. Through this process, health care workers have built more intimate relationships and rapport with residents. The Culture Change Movement of nursing homes in the past few decades has long emphasized the promotion of person-centered and holistic care in residential care homes [52,53], and the experiences of health care workers in this study indicate the valuable contributions of person-centered and holistic forms of care in tackling complex health care challenges brought about by the pandemic. In addition, while existing studies often considered that psychosocial care involved a high stake of emotional labor that could be stressful for care workers [54,55,56], our study suggests that building genuine bonding with older people does not only maximize older people’s quality of life in restrictive circumstances, but also brings a sense of meaningfulness and job satisfaction to health care workers. In the future, the LTC sector should nurture health care workers’ roles and capacity to attend to the psychosocial wellbeing of older people through evidence-based support, including enhancing supervisor connection and communication and initiating peer support programs [54].

The findings of this study shed light on the impacts of the COVID-19 pandemic on the health care workforce in the residential aged care sector. This study found mixed results about the long-term impacts of the pandemic on the care workforce in residential care homes. Existing research predominantly focuses on the adversities that health care workers have faced during the COVID-19 pandemic, such as acute workforce shortages, anxiety, stress, and physical and emotional burnout [10,14,16,57,58]. While this study also found detrimental impacts on health care workers, it proposes that health care workers were able to adapt and seek resilience in adversity over time, such as by improvising new modes of care delivery to meet residents’ needs during the pandemic. Although IPC precautionary measures have escalated the workforce crisis in long-term care facilities, this study witnessed new health care workers joining the aged care sector because of the relative job security it offers, as well as the continuation of positive work experiences and satisfaction among health care workers during the pandemic. These findings offer new insights into the discussion of the chronic direct care workforce deficit in the LTC sector [59,60,61]. In the long run, the LTC sector will benefit from workforce development and support initiatives that nurture the resilience and value the on-the-ground knowledge and experiences of frontline health care workers.

Lastly, this study has broader implications for the delivery of care to older people in LTC facilities in other contexts beyond Hong Kong. The main themes discussed in the study regarding enhancing IPC measures, maintaining the psychosocial wellbeing of residents, and developing resilience can be applied across sociocultural contexts. Health care workers and LTC facilities in different countries have been facing similar long-term challenges of the COVID-19 pandemic in their delivery of care. The innovative modes of care delivery in the context of Hong Kong, such as redesigning age-friendly IPC measures, strengthening the roles of health care workers in psychosocial care, and nurturing resilience in adverse circumstances, also engender new insights for other countries to better adapt to the persistent impacts of the COVID-19 pandemic. Moreover, the current study indicates the necessity of culture-sensitive care in LTC facilities to cope with the adversities brought by the COVID-19 pandemic. The implication of maintaining intergenerational bonding in the pandemic can be meaningful for other East Asian countries that share similar sociocultural backgrounds of strong intergenerational ties, as well as for LTC facilities in the West that serve East Asians in their diaspora.

## 5. Conclusions

The present study examined the long-term impacts of the COVID-19 pandemic on residential care homes from the perspectives of health care workers in Hong Kong. During the pandemic, IPC measures became an essential part of the daily care routines of health care workers in residential care homes. While the COVID-19 pandemic has had detrimental impacts on the psychosocial wellbeing of residents, it has also raised health care workers’ awareness of and practices related to psychosocial care, including building rapport with older people and sustaining intergenerational bonding. In the midst of difficult circumstances, although health care workers encountered numerous challenges, especially at the onset of the pandemic, they developed resilience and adapted to transforming ways of delivering care over time.

## Figures and Tables

**Table 1 ijerph-19-15287-t001:** Demographic characteristics of participants (*n* = 30).

Characteristic	Number of Participants (*n* = 30)
Role	
Registered Nurse	5
Enrolled Nurse	8
Health Worker	8
Personal Care Worker	9
Sex	
Male	5
Female	25
Education	
Primary and below	1
Secondary	13
Post-secondary	16
Age (years old)	
18–25	5
26–44	15
45–59	9
60 or above	1
Affiliated residential care homes	
Subvented	6
Contract (provide both subvented and non-subvented places)	16
Private	8

**Table 2 ijerph-19-15287-t002:** Characteristics of residential care homes (*n* = 6).

Facility Number	Funding Nature	Number of Beds	Number of Health Care Workers **	Health Care Workers-to-Beds Ratio
1	Private	325	82	0.25
2	Contract *	102	65	0.63
3	Contract *	114	60	0.53
4	Contract *	147	76	0.51
5	Contract *	150	N.A. ***	N.A.
6	Public	109	51	0.47

* Contract facilities provide both publicly subsidized and privately funded beds. ** The number of health care workers refers to the number of full-time and part-time staff working in the roles of registered nurses, enrolled nurses, health workers, and personal care workers. *** The number is not disclosed by the residential care home.

**Table 3 ijerph-19-15287-t003:** Themes and sub-themes that emerged from the data.

Themes	Sub-Themes
Enhancing infection prevention and control measures	(1) Incorporating IPC measures in daily care routines
	(2) Performing health education and promotion
2. Maintaining the psychosocial wellbeing of residents	(1) Interruption of intergenerational bonding
	(2) Witnessing the adverse wellbeing outcomes of residents
	(3) Sustaining intergenerational bonding in adaptive ways
	(4) Building rapport with residents
3. Developing resilience	(1) Experiencing stress at the onset of the pandemic
	(2) Seeking resilience in adverse circumstances
	(3) Joining the LTC sector in the midst of the pandemic

## Data Availability

Not applicable.

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
