# Peer review of "Caring for Older People during and beyond the COVID-19 Pandemic: Experiences of Residential Health Care Workers"

_ijerph, 2022, doi:10.3390/ijerph192215287_

Round 1

Reviewer 1 Report

In the manuscript entitled "Caring for older people during and beyond the COVID-19 pandemic: Experiences of residential health care workers”, the authors studied the impacts of theCOVID-19 pandemic on the delivery of care in residential care homes from the perspectives of health care workers.

This qualitative study analysed the experiences and perception of health care workers from 6 residential care homes in Hong Kong. Thirty health care workers were interviewed and answered guided questions including:

1. As the COVID-19 pandemic has been ongoing for more than two years, from your experiences, what are the impacts of the pandemic on the delivery of care in your residential care home?

2. In what ways do you think the delivery of care during the pandemic differs from that in the past?

3. How do you and your colleagues cope with and adapt to these changes?

4. From your perspective, what are the impacts of these changes on the quality of life of residents?

The authors analysed the text of the interviews using thematic analysis. In the manuscript, they nicely illustrated the answer to these questions using part of the interview. This study has a great potential but need to be completed.

1) In order to make the article easier to read, the authors should write in italic the sentences coming from the interviews.

2) Line 396: paragraph number “4.3.1 Experiencing stress at the onset of the pandemic “ should be replaced by 3.3.1.

3) In material and methods, line 180-182, the authors mentioned that “participants were purposively recruited from different residential care homes, including publicly subsidized and private facilities, to allow for heterogeneous work experiences and perspectives among health care workers working on the frontline during the COVID-19 pandemic”. Based on this sampling strategy, did the authors find difference between experiences of health care workers from public and private residential care homes?

4) The authors recruited only 5 men health care workers. Were their experiences different from those of the women health care workers?

5) As mentioned in table 1, the author also recorded experiences from health care workers with different roles (Registered Nurse, Enrolled Nurse, Health Worker, Personal Care Worker). Did the author find difference in lived experience depending on the role of the health care workers?

6) In table 1, it will be interesting to add facility characteristics such as type of room (private or shared), size of the facilities (number of residents). These informations could have an impact on the quality of life of the residents. Could the authors complete table 1 and comment this point.

7) One of the questions asked to the health care workers was “From your perspective, what are the impacts of these changes on the quality of life of residents?”. How did health care workers evaluate the quality of life of the residents? The authors showed how the quality of life of the resident decrease with the restrictions but did health care workers witness improvement in quality of life after planning activities and coordinating video call or visiting arrangement. Furthermore, did the authors obtain informations that improving the quality of life of the residents would increase the well-being of employees and vice-versa?

8) The COVID-19 pandemic was a particularly stressful situation for health care workers and patients. Nevertheless, the authors only briefly mentioned this aspect. The authors should emphasize the adversities and stress that health care workers have faced in part 3.3.1. they could use the following exemple:

Line 248: “I worried I might infect the residents.”

Line 400: “In the beginning, I was quite worried because our institution was quite big and busy. So, I worry [about being infected].”

Line 436: “As health care workers, we also avoid going to public places as much as possible [to avoid infection]”.

9) In the discussion, could the authors make recommendations in terms of how to work and take care of older people in residential care homes after COVID-19 pandemic?

Author Response

Dear Reviewer, 

Thank you so much for offering us the opportunity to submit a revised version of the manuscript “Caring for Older People during and beyond the COVID-19 Pandemic: Experiences of Residential Health Care Workers”. We appreciate your valuable input on our paper. Your thoughtful comments brought in many new and important insights that help improve the current version of our manuscript. We have carefully considered the comments and tried our best to address all of them. Please find the point-by-point details of the revisions and our responses below. All modifications in the manuscript were marked up as “track changes” in the manuscript. Again, we appreciate your valuable comments a lot!

Comment 1. In order to make the article easier to read, the authors should write in italic the sentences coming from the interviews.

Response 1. All the quotations were italicized to facilitate easier reading. Thank you so much for this suggestion!

Comment 2. Line 396: paragraph number “4.3.1 Experiencing stress at the onset of the pandemic “ should be replaced by 3.3.1.

Response 2. Thank you for noticing this error! We have corrected the number.

Comment 3. In material and methods, line 180-182, the authors mentioned that “participants were purposively recruited from different residential care homes, including publicly subsidized and private facilities, to allow for heterogeneous work experiences and perspectives among health care workers working on the frontline during the COVID-19 pandemic”. Based on this sampling strategy, did the authors find difference between experiences of health care workers from public and private residential care homes?

Response 3. We have noticed that the difference in health care workers’ experiences among different residential care homes manifested in two ways, which were added to the manuscript. First, since private residential care homes often have lower health care workers-to-beds ratio in our study and Hong Kong in general, health care workers in the private home experienced more challenges when they had to cope with the increased workloads caused by IPC measures directly and indirectly (lines 409-412). Secondly, when discussed that several personal care workers joined the long-term care sector during the pandemic, public facilities were more attractive to them because of the better job security this type of home offered(lines 745-755).

Comment 4. The authors recruited only 5 men health care workers. Were their experiences different from those of the women health care workers?

Response 4. The gender ratio of research participants we recruited generally reflected the imbalanced gender ratio of the workforce in the LTC sector, for care has long been socially constructed as a highly gendered domain and provided predominantly by female labor. That said, this study has the limitation that it didn’t recruit a larger number of male participants. From the data we got, we couldn’t draw a convincing conclusion about the gender differences in health care workers’ care experiences. We believed that their differences had more to do with their roles than genders.

Comment 5. As mentioned in table 1, the author also recorded experiences from health care workers with different roles (Registered Nurse, Enrolled Nurse, Health Worker, Personal Care Worker). Did the author find difference in lived experience depending on the role of the health care workers?

Response 5. Thank you so much for pointing this out! Yes, we added the analysis of their roles. We found that personal care workers’ (PCWs’) care workloads were particularly affected by the IPC measures because they undertake the majority of direct care tasks that used to be shared by family caregivers before the pandemic (lines 375-409; 802-808). Besides, we found that several participants joined the LTC sector as PCWs after losing their jobs in other sectors during the pandemic. These PCWs were particularly drawn into the relatively adequate job opportunities and good job security of the sector. We further discussed PCWs’ perspectives on retaining in the LTC sector after the pandemic (lines 745-755). We also found that health care workers in other roles generally played a more important role in communicating with family members (lines 558-560).

Comment 6. In table 1, it will be interesting to add facility characteristics such as type of room (private or shared), size of the facilities (number of residents). These informations could have an impact on the quality of life of the residents. Could the authors complete table 1 and comment this point.

Response 6. We added table 2 (lines 297-303) to elaborate on the facility characteristics in detail.

Comment 7. One of the questions asked to the health care workers was “From your perspective, what are the impacts of these changes on the quality of life of residents?”. How did health care workers evaluate the quality of life of the residents? The authors showed how the quality of life of the resident decrease with the restrictions but did health care workers witness improvement in quality of life after planning activities and coordinating video call or visiting arrangement. Furthermore, did the authors obtain informations that improving the quality of life of the residents would increase the well-being of employees and vice-versa?

Response 7. Again, thank you so much for such an insightful comment! Yes, we extended these findings by elaborating on how health care workers used adaptive ways to sustain the intergenerational bonding of older adults and relax their emotional distress (lines 554-557). We also further elaborated that the new modes of care provision could ensure the continuity of care to the extent possible and was pivotal to the quality of life of older people during the pandemic. This process promoted job satisfaction among health care workers (lines 714-733).

Comment 8. The COVID-19 pandemic was a particularly stressful situation for health care workers and patients. Nevertheless, the authors only briefly mentioned this aspect. The authors should emphasize the adversities and stress that health care workers have faced in part 3.3.1. they could use the following exemple:

Line 248: “I worried I might infect the residents.”

Line 400: “In the beginning, I was quite worried because our institution was quite big and busy. So, I worry [about being infected].”

 Line 436: “As health care workers, we also avoid going to public places as much as possible [to avoid infection]”.

Response 8: The stressful experiences of health care workers were further elaborated (lines 665-676).

Comment 9. In the discussion, could the authors make recommendations in terms of how to work and take care of older people in residential care homes after COVID-19 pandemic?

Response 9. We discussed the recommendations for caring for older people in LTC facilities after the COVID-19 pandemic at the end of each paragraph in the discussion section. Each paragraph discussed one theme of the implications of the study and recommendations were added in response to each theme (lines 804-808; 825-827; 912-916; 930-933; 951-954989-991).

Again, thank you so much for your thoughtful review and comments.

Reviewer 2 Report

2.In the Results 216

In the Results section, describe only the results of this survey. Do not write your reflections based on references from the literature or your own experience. Similarly, the fact that public health policy influences the results of the survey is a prerequisite for your research and should not be included in the results section. This is because many of the questions in this survey are open questions. In the results section, it is not appropriate for the author to explain the premise of the question. The following is an example of what is easy to understand, so please correct the other parts of the "Results" as well.

Following the guidelines of the Center for Health Protection of the Hong Kong government, residential care homes are required to take a 234 number of precautionary measures to minimize the risk of contracting and spreading 235 COVID-19 [30].”  

These statements 234-235 are not results, so please describe them in the discussion section.

While the implementation of IPC measures have now been directly integrated into 254 the daily care routine of health care workers, these preventive measures also unintentionally and indirectly affected other aspects of care delivery in residential care homes. 256

This is particularly the case of the implementation of visiting restriction policy.“

Please clarify whether the statements in 254-256 were concluded (1) directly from the results of the survey, (2) based on citations from the literature, etc., or (3) including the author's experience. In the case of (2) or (3), please move this statement to the discussion section.

Although the visiting restriction 259 was lifted to allow family members to visit residents in person in the form of scheduled 260 sessions, family caregivers were still banned from visiting the residents in the living area 261 and the frequency and length of visits were restricted compared to the pre-pandemic policy. In the face of the prolonged visiting restriction policy,” .

This description of 259-261 is not the result of the investigation. These should be described in the discussion section.

During 257 the pandemic, residential care homes in Hong Kong have imposed new visiting policies 258 that restrict family members from visiting the residents.”

This description of 257-258 is not the result of the investigation. These should be described in the discussion section.

3.2 Maintaining the psychosocial wellbeing of residents 302

 When discussing the long-term impacts of the pandemic, most health care workers 303 suggested that the most significant impact was on the psychosocial wellbeing of residents in residential care homes. Health care workers revealed that IPC measures, especially visiting restriction, had disrupted the intergenerational bonding between older people and their family members and brought adverse wellbeing outcomes to older people. In order to tackle these challenges, health care workers had developed adaptive 308 ways of sustaining the intergenerational bonding and building rapport with residents. 309 “

Please clarify whether the statements in 302-308 were (1) drawn directly from the results of the survey, (2) were concluded based on references from the literature, etc., or (3) included the author's experience. In the case of (2) or (3), please move this statement to the discussion section.

Author Response

Dear Reviewer,

Thank you so much for offering us the opportunity to submit a revised version of the manuscript “Caring for Older People during and beyond the COVID-19 Pandemic: Experiences of Residential Health Care Workers”. We appreciate your valuable input on our paper. Your thoughtful comments brought in many new and important insights that help improve the current version of our manuscript. We have carefully considered the comments and tried our best to address all of them. Please find the point-by-point details of the revisions and our responses below. All modifications in the manuscript were marked up as “track changes” in the manuscript. Again, we appreciate your valuable comments a lot!

  1. Thank you so much for pointing out that it is inappropriate to explain the premise of the questions in the Result section. We have carefully re-read the Result section and deleted those sentences you carefully indicated in the comments. We also clarify other places we found problematic by making sure that the result session discussed only direct findings from the interviews (e.g. lines 346, 434-435, 477, 495). Hopefully, the Result section is now serving the right purpose.
  2. The statement, “While the implementation of IPC measures have now been directly integrated into the daily care routine of health care workers, these preventive measures also unintentionally and indirectly affected other aspects of care delivery in residential care homes.” came from our summarization of the participants’ experiences and thus was kept in the manuscript.
  3. The statement, “3.2 Maintaining the psychosocial wellbeing of residents. When discussing the long-term impacts of the pandemic, most health care workers suggested that the most significant impact was on the psychosocial wellbeing of residents in residential care homes. Health care workers revealed that IPC measures, especially visiting restrictions, had disrupted the intergenerational bonding between older people and their family members and brought adverse wellbeing outcomes to older people. In order to tackle these challenges, health care workers had developed adaptive ways of sustaining intergenerational bonding and building rapport with residents.”, also came from the participants’ experiences and thus was kept in the manuscript.

Again, thank you so much for your thoughtful review and comments.

Reviewer 3 Report

The manuscript reports on a qualitative study of healthcare professionals' perception of how the COVID-19 pandemic impacted routine in long-term care settings. The originality and significance of this study is hard to judge given the lack of a clear positioning (see comments below).

Also, the results section needs a serious rework in the form of unpacking, interpreting, and analyzing the presented quotes.

Last, but not least, the discussion section needs to engage more with relevant literature and take a slightly broader perspective.

Abstract:

* "semi-structured" instead of "semi-structural".

* What is "clear infection prevention and control planning"?

* what is "human-centered and holistic care"?

* you are promising a lot of outcomes at the end of the abstract. can your findings deliver on all of them? if yes, this needs to be clearer.

Introduction:

* Why is the need for care social here? Maybe "societal need" rather than "social need"?

* Line 125: You write that studies like yours are "scarce". This implies that there are other studies that should be cited here.

* Line 195: How can the pandemic have been on-going for two years in February 2021?

* Lines 195-206: these 4 points are very general and overlapping to a large degree. to what degree did you rely on these questions? what degree of freedom did you employ in asking further/different questions?

* Lines 207-2015: This is a very short data analysis section for such a type of study. Was the process entirely inductive? Was the same material coded by at least two researchers? Etc. pp.

* Line 249: Maybe write "informant 20" or "participant 20" rather than "case 20".

Results

* Section 3.3.3 (as an example - other subsections have similar problems): 16 lines of verbatim quotes from interviews cannot be interpreted and anlalyzed in 8 lines of mostly introductory text. You need (in general in the Results section) to provide interpretation and analysis - not leave this up to the reader.

Discussion:

* generally, you need to engage more with the literature than to just say that your results are in line with existing ones. surely, there are differences and nuances here?

* this section would profit from relating to existing work on the lived experiences of frontline staff during the COVID-19 pandemic in long-term care settings: https://doi.org/10.1016/j.ajic.2021.03.006

* this section would also profit from engaging with other work on COVID-19 pandemic, frontline staff, and their relations to relatives. for, example consider work on how schools caring for dependent relatives (here children, not elderly) finds similar needs for clear communication etc.: https://doi.org/10.1080/13698575.2022.2028743

* generally, you come in on communication during the findings - but this is entirely absent from the discussion

Author Response

Dear Reviewer,

Thank you so much for offering us the opportunity to submit a revised version of the manuscript “Caring for Older People during and beyond the COVID-19 Pandemic: Experiences of Residential Health Care Workers”. We appreciate your valuable input on our paper. Your thoughtful comments brought in many new and important insights that help improve the current version of our manuscript. We have carefully considered the comments and tried our best to address all of them. Please find the point-by-point details of the revisions and our responses below. All modifications in the manuscript were marked up as “track changes” in the manuscript. Again, we appreciate your valuable comments a lot!

Comment 1: The manuscript reports on a qualitative study of healthcare professionals' perception of how the COVID-19 pandemic impacted routine in long-term care settings. The originality and significance of this study is hard to judge given the lack of a clear positioning (see comments below).

 Also, the results section needs a serious rework in the form of unpacking, interpreting, and analyzing the presented quotes.

Last, but not least, the discussion section needs to engage more with relevant literature and take a slightly broader perspective.

Response 1: Thank you so much for indicating the limitations of the manuscript! We have carefully reviewed the manuscript and made amendments based on your suggestions. The changes will be further elaborated on below.

Abstract section:

Comment 2. "semi-structured" instead of "semi-structural".

Response 2. Thank you for pointing out this error! We have corrected the phrase to “semi-structured interview”. (line 18).

Comment 3.  What is "clear infection prevention and control planning"?

 what is "human-centered and holistic care"?

you are promising a lot of outcomes at the end of the abstract. can your findings deliver on all of them? if yes, this needs to be clearer.

Response 3. Because of the word limit of the Abstract section, we didn’t elaborate in detail regarding the meaning of all these terms, and this might cause confusion to readers. In order to make the abstract section clear and concise, we deleted the sentence that discussed these implications and only elaborate on them in the Discussion section. We hope that the abstract will look much clearer now (lines 22-23).

Introduction section:

Comment 4. Why is the need for care social here? Maybe "societal need" rather than "social need"?

Response 4. We amended it to “need”. (line 28).

Comment 5. Line 125: You write that studies like yours are "scarce". This implies that there are other studies that should be cited here.

Response 5. We added a few citations of other in-depth qualitative studies that investigated the impacts of the pandemic on the delivery of care in long-term care (LTC) facilities (line 157).

 Comment 6. Line 195: How can the pandemic have been on-going for two years in February 2021?

Response 6. Thank you so much for indicating this error! We have amended it to “more than one year” (line 231).

Comment 7. Lines 195-206: these 4 points are very general and overlapping to a large degree. to what degree did you rely on these questions? what degree of freedom did you employ in asking further/different questions?

Response 7. Semi-structured interview highlights relational focus and flexibility in the data collection process. The four questions served primarily as a flexible interview guide for us to explore and delve into research participants’ experiences and perspectives of care delivery during the pandemic. The four questions also provide us guidance to purposively ask follow-up questions based on participants’ responses. Our intentions of using the four questions were in exploring participants’ experiences from different dimensions covered by these questions (i.e. the general impacts of the pandemic, comparing delivery of care during and before the pandemic in greater detail, the coping and adaptation of health care workers, and the impacts on the quality of life of older people). We hope these illustrations could resolve your concerns.

Comment 8. Lines 207-2015: This is a very short data analysis section for such a type of study. Was the process entirely inductive? Was the same material coded by at least two researchers? Etc. pp.

Response 8. We extended the data analysis section by describing the roles of all authors in the process of thematic analysis (line 266-278).

Comment 9. Line 249: Maybe write "informant 20" or "participant 20" rather than "case 20".

Response 9. We used “participant” for all the quotations. Thank you so much for indicating this, it is a very important point indeed.

Result section:

Comment 10. * Section 3.3.3 (as an example - other subsections have similar problems): 16 lines of verbatim quotes from interviews cannot be interpreted and anlalyzed in 8 lines of mostly introductory text. You need (in general in the Results section) to provide interpretation and analysis - not leave this up to the reader.

Response 10. We have reworked the Results section thoroughly and tried to unpack the quotes as much as possible (lines 375-412; 445-453;487-493; 502-511; 543-546; 554-577; 665-676;714-733; 745-755).

Discussion section:

Comment 11. generally, you need to engage more with the literature than to just say that your results are in line with existing ones. surely, there are differences and nuances here?

Response 11. We totally agree with you that we need to indicate what our study added to existing literature more clearly. In each of the themes in the discussion section, we reframed or added the differences and nuances of the current study (lines 809-823; 836-844; 926-930; 936-939; 946-951; 958-989).

Comment 12. this section would profit from relating to existing work on the lived experiences of frontline staff during the COVID-19 pandemic in long-term care settings: https://doi.org/10.1016/j.ajic.2021.03.006

Response 12. Absolutely! We cited this paper as well as other existing work on the lived experiences of frontline staff in this section (line 796).

Comment 13. this section would also profit from engaging with other work on COVID-19 pandemic, frontline staff, and their relations to relatives. for, example consider work on how schools caring for dependent relatives (here children, not elderly) finds similar needs for clear communication etc.: https://doi.org/10.1080/13698575.2022.2028743

Generally, you come in on communication during the findings - but this is entirely absent from the discussion

Response 13. Thank you so much for this insightful comment! Communication is a very important point indeed. We learned a lot from the paper you recommended. We added this paper and other work to further discuss the issue of communication between health care workers and family members in a new paragraph (line 923).

Thank you so much for your review and comment! 

Reviewer 4 Report

An excellent paper on an interesting and timely piece of research. Only thing I would add is some additional discussion as to the relevance of the research on similar settings elsewhere globally - can the findings help in other countries etc.?

Author Response

Dear Reviewer,

Thank you so much for offering us the opportunity to submit a revised version of the manuscript “Caring for Older People during and beyond the COVID-19 Pandemic: Experiences of Residential Health Care Workers”. We appreciate your valuable input on our paper. Your thoughtful comments brought in many new and important insights that help improve the current version of our manuscript. We have carefully considered the comments and tried our best to address all of them. Please find the point-by-point details of the revisions and our responses below. All modifications in the manuscript were marked up as “track changes” in the manuscript. Again, we appreciate your valuable comments a lot!

Comment 1. An excellent paper on an interesting and timely piece of research. Only thing I would add is some additional discussion as to the relevance of the research on similar settings elsewhere globally - can the findings help in other countries etc.?

Response 1. Thank you so much for your insightful comments! We totally agree that we need to extend the implications of this study to other countries. We added a paragraph to elaborate on this point (lines 992-1006). Hopefully, this will make our findings more relevant to readers from other sociocultural contexts.

Again, thank you so much for your thoughtful review and comments!

Round 2

Reviewer 3 Report

The authors have successfully addressed all the issues from the first round. As a consequence, the manuscript stand much clearer and with a stronger contribution.

The answer to our comment 7 (see below for reference) indeed helps to alleviate my concerns. The manuscript would, however, profit from amending that description with a 1 or max 2 sentences summarizing answer 7.

FOR REFERENCE:

Comment 7. Lines 195-206: these 4 points are very general and overlapping to a large degree. to what degree did you rely on these questions? what degree of freedom did you employ in asking further/different questions?

Response 7. Semi-structured interview highlights relational focus and flexibility in the data collection process. The four questions served primarily as a flexible interview guide for us to explore and delve into research participants’ experiences and perspectives of care delivery during the pandemic. The four questions also provide us guidance to purposively ask follow-up questions based on participants’ responses. Our intentions of using the four questions were in exploring participants’ experiences from different dimensions covered by these questions (i.e. the general impacts of the pandemic, comparing delivery of care during and before the pandemic in greater detail, the coping and adaptation of health care workers, and the impacts on the quality of life of older people). We hope these illustrations could resolve your concerns.

Author Response

Dear Reviewer,

Thank you so much for your prompt feedback! We really appreciate it.

Based on your comment, we added two sentences to elaborate on the use of guiding questions in the semi-structured interviews (Lines 239-244). Again, thank you so much for taking the time and effort to provide thoughtful comments on this manuscript!
